# Transcriptomic profiling and genomic mutational analysis of Human coronavirus (HCoV)-229E -infected human cells

Nehemya Friedman[1,2], Jasmine Jacob-Hirsch[3,4], Yaron Drori[1,2], Eyal Eran[3,4], Nitzan Kol[3,4], Omri Nayshool[3,4], Ella Mendelson[1,2], Gideon Rechavi[3,4], Michal Mandelboim[1,2]*

**1** Central Virology Laboratory, Ministry of Health, Chaim Sheba Medical Center, Tel-Hashomer, Ramat Gan, Israel, **2** Department of Epidemiology and Preventive Medicine, School of Public Health, Sackler Faculty of Medicine, Tel-Aviv University, Tel-Aviv, Israel, **3** Sheba Cancer Research Center (SCRC), Chaim Sheba Medical Center, Ramat Gan, Israel, **4** Wohl Centre for Translational Medicine, Chaim Sheba Medical Center, Ramat Gan, Israel

* michalman@sheba.health.gov.il

**Data Availability Statement:** RNA-Seq library is available in the GEO repository (Accession number GSE155986).

## Abstract

Human coronaviruses (HCoVs) cause mild to severe respiratory infection. Most of the common cold illnesses are caused by one of four HCoVs, namely HCoV-229E, HCoV-NL63, HCoV-HKU1 and HCoV-OC43. Several studies have applied global transcriptomic methods to understand host responses to HCoV infection, with most studies focusing on the pandemic severe acute respiratory syndrome coronavirus (SARS-CoV), Middle East respiratory syndrome CoV (MERS-CoV) and the newly emerging SARS-CoV-2. In this study, Next Generation Sequencing was used to gain new insights into cellular transcriptomic changes elicited by alphacoronavirus HCoV-229E. HCoV-229E-infected MRC-5 cells showed marked downregulation of superpathway of cholesterol biosynthesis and eIF2 signaling pathways. Moreover, upregulation of cyclins, cell cycle control of chromosomal replication, and the role of BRCA1 in DNA damage response, alongside downregulation of the cell cycle G1/S checkpoint, suggest that HCoV-229E may favors S phase for viral infection. Intriguingly, a significant portion of key factors of cell innate immunity, interferon-stimulated genes (ISGs) and other transcripts of early antiviral response genes were downregulated early in HCoV-229E infection. On the other hand, early upregulation of the antiviral response factor Apolipoprotein B mRNA editing enzyme catalytic subunit 3*B (APOBEC3B)* was observed. APOBEC3B cytidine deaminase signature (C-to-T) was previously observed in genomic analysis of SARS-CoV-2 but not HCoV-229E. Higher levels of C-to-T mutations were found in countries with high mortality rates caused by SARS-CoV-2. APOBEC activity could be a marker for new emerging CoVs. This study will enhance our understanding of commonly circulating HCoVs and hopefully provide critical information about still-emerging coronaviruses.

**Funding:** The authors received no specific funding for this work.

**Competing interests:** The authors have declared that no competing interests exist.

## Introduction

Human coronaviruses (HCoVs), which are enveloped, positive single-stranded RNA (+-ssRNA) viruses belonging to the *Coronaviridae* family, are associated with a wide spectrum of respiratory diseases [1, 2]. Thus far, seven types of HCoV have been discovered in humans, the majority (Alphacoronavirus species HCoV-NL63 and HCoV-229E and betacoronavirus species HCoV-HKU1 and HCoV-OC43) being causative agents of community-acquired infections. These HCoV infections occur primarily in the winter-spring season [3–6], however summer epidemics have been detected [7]. The other three HCoVs are the pandemic severe acute respiratory syndrome coronavirus (SARS-CoV), the Middle East respiratory syndrome coronavirus (MERS-CoV) [8] and the newly pandemic SARS-CoV-2 [9].

HCoVs typically exploit three known receptors or O-acetylated sialic acids on the host cell surface to gain entry [10]. Coronaviruses have an exceptionally large ~30 kb genome in polycistronic organization and engage a unique transcription mechanism to generate mRNA [11]. They possess many non-structural proteins (NSPs) that appear to have multiple functions in RNA processing, facilitating viral entry, gene expression and virus release [10]. Many +ssRNA viruses use diverse strategies to subvert and exploit host protein synthesis [12], with one central strategy involving inactivation of the elongation initiation complex by manipulating kinase activities. Indeed, viral replication and protein expression are associated with endoplasmic reticulum (ER) overload and subsequent stress responses which trigger host protein translation shutoff. Coronaviruses have evolved to block or subvert these responses in order to restore ER homeostasis or block apoptosis [13]. Some coronaviruses have evolved to counteract the dsRNA-activated protein kinase (PKR), which prevents translation by phosphorylating elongation initiation factor 2 (eIF2) and upregulating antiviral gene expression, including the production of interferons (IFNs) [10, 14]. Several additional innate immune receptors involved in recognizing RNA viruses activate interferon-stimulated genes (ISGs) and other cellular factors which fight the viral intruders [15].

In the last decade, several studies have focused on host cell transcriptome changes subsequent to HCoV infection. Microarray analysis in human A549 and HuH7 cells found HCoV-229E to attenuate the inducible activity of NF-κB and to restrict the nuclear concentration of NF-κB subunits [16]. Another microarray analysis of peripheral blood mononuclear cells collected from SARS-CoV patients, described the altered inflammatory and immune genes that may contribute to acute lung injury and imbalance of homeostasis [17]. Differential RNA sequencing analysis comparing SARS-CoV and MERS-CoV–infected human Calu-3 cells showed that both viruses induced a similar activation pattern of recognition receptors and of the interleukin 17 (IL-17) pathway, however, only MERS-CoV downregulated the expression of type I and II major histocompatibility complex genes [18]. Transcriptomic characteristics of SARS-CoV-2 patients identified excessive cytokine release, activation of apoptosis and of the p53 signaling pathway [19]. Global transcriptomic sequencing to shed light on the influence of common circulating HCoVs on host cells is lacking. Thus, this study used RNA-next generation sequencing (NGS) to track the transcriptomic changes in human MRC-5 cells following infection with HCoV-229E.

## Results

RNA was extracted at various time points following infection of MRC-5 with HCoV-229E and viral titers were estimated both from the infected cells as well as the from culture supernatants by qPCR analysis (Fig 1A and 1B). Rapid viral propagation inside the host cells, reaches maximal levels at 24 hours post infection (HPI) and remained stable for 4 days (Fig 1A). Viral copies began to emerge in the supernatant at 24 HPI and gradually increased over time (Fig 1B).

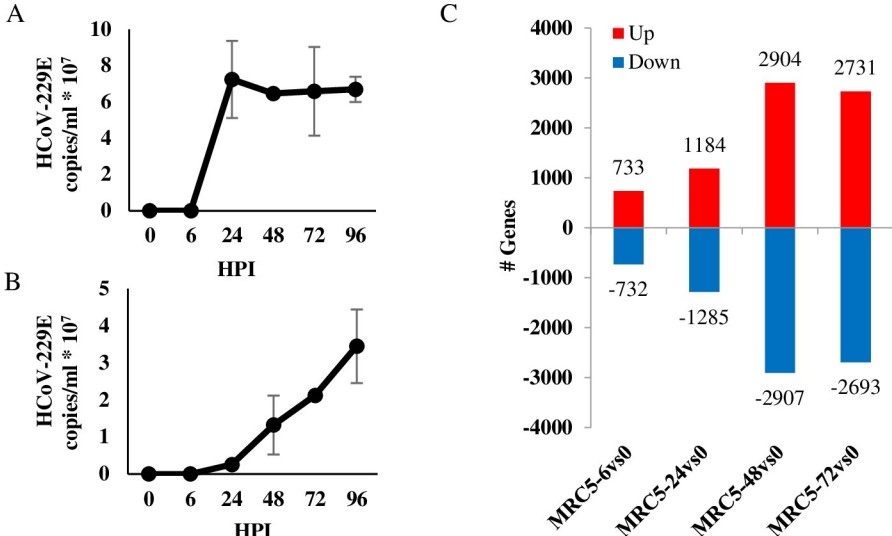

**Fig 1. Change in gene expression following HCoV-229E infection.** (A-B) Viral propagation of HCoV-229E. MRC-5 cells, 24-hour culture to 80–90% confluency, were infected with 0.01 multiplicity of infection (MOI) of HCoV-229E virus. Cells and virus were grown in Eagle's Minimum Essential Medium (EMEM) supplemented with 2% fetal calf serum (FCS) and incubated in a humidified incubator with 5% $CO_2$, at 35˚C. Total RNA was extracted at the indicated hours post infection (HPI) and viral load was determined in both cells (A) or growth media (B) by qPCR analysis. (C) Active transcripts in HCoV-229E –infected MRC-5 cells. Total RNA samples of MRC-5-infected cells (from section A) were prepared using the Illumina TruSeq RNA Library Preparation Kit v2 and sequenced in Illumina HiSeq 2500 Sequencer. Software and applications used for sequencing and data analysis are specified in the methods. Total number of upregulated (>2 fold change, red bars) or downregulated (<-2 fold change, blue bars) genes at the indicated HPI was calculated by comparing to gene expression to that measured in uninfected cells.

At 96 HPI, MRC-5 cell counts began to decline and the cells were not viable a day later. In parallel to the high replication rate of HCoV-229E, a robust change in the MRC-5 transcriptome was observed, with 1465 transcripts already showing a >2-fold change at 6 HPI ('early genes') as compared to uninfected cells. The number of active genes continued to increase as infection progressed, until it reached over 5000 transcripts at 48–72 HPI (Fig 1C).

Prediction of canonical pathways based on the most significant gene enrichment included eIF2 signaling, cell cycle control of chromosomal replication, cholesterol biosynthesis super-pathway, BRCA1-associated DNA damage response and other cell cycle checkpoint regulation pathways (Fig 2A). More specifically, 205 gene annotations in the NGS analysis were categorized in the eIF2 signaling pathway, of which 64.88% genes were active/inactive in MRC-5 cells. The parameter of active/inactive transcript is being significantly >2 upregulated or <-2 downregulated in fold-change as compared to uninfected cells. At early stages of infection, considerable downregulation of the eIF2 signaling pathway was observed in MRC-5 cells, a trend which increased as infection progressed (Fig 2B-I). In this pathway, over 80 genes which code for ribosomal protein subunits (RPLs and RPSs) were downregulated during infection (data not presented). The cell cycle control of chromosomal replication pathway includes genes that are involved in the strict control of DNA origin replication points. In our analysis, 57 genes were annotated to this pathway in IPA software, of which 64.91% were active/inactive in MRC-5 cells. Genes in this pathway were upregulated mostly at the early stages of HCoV-229E infection (22 genes at 6 HPI), but were later less active during the late stages of infections (Fig 2B-II). Two of the most significantly upregulated early genes in this pathway were ORC1 and CDC6, both of which initiate assembly of a pre-replication complex at DNA replication origins [20]. In the cholesterol biosynthesis pathway, 18 genes were downregulated in MRC-5

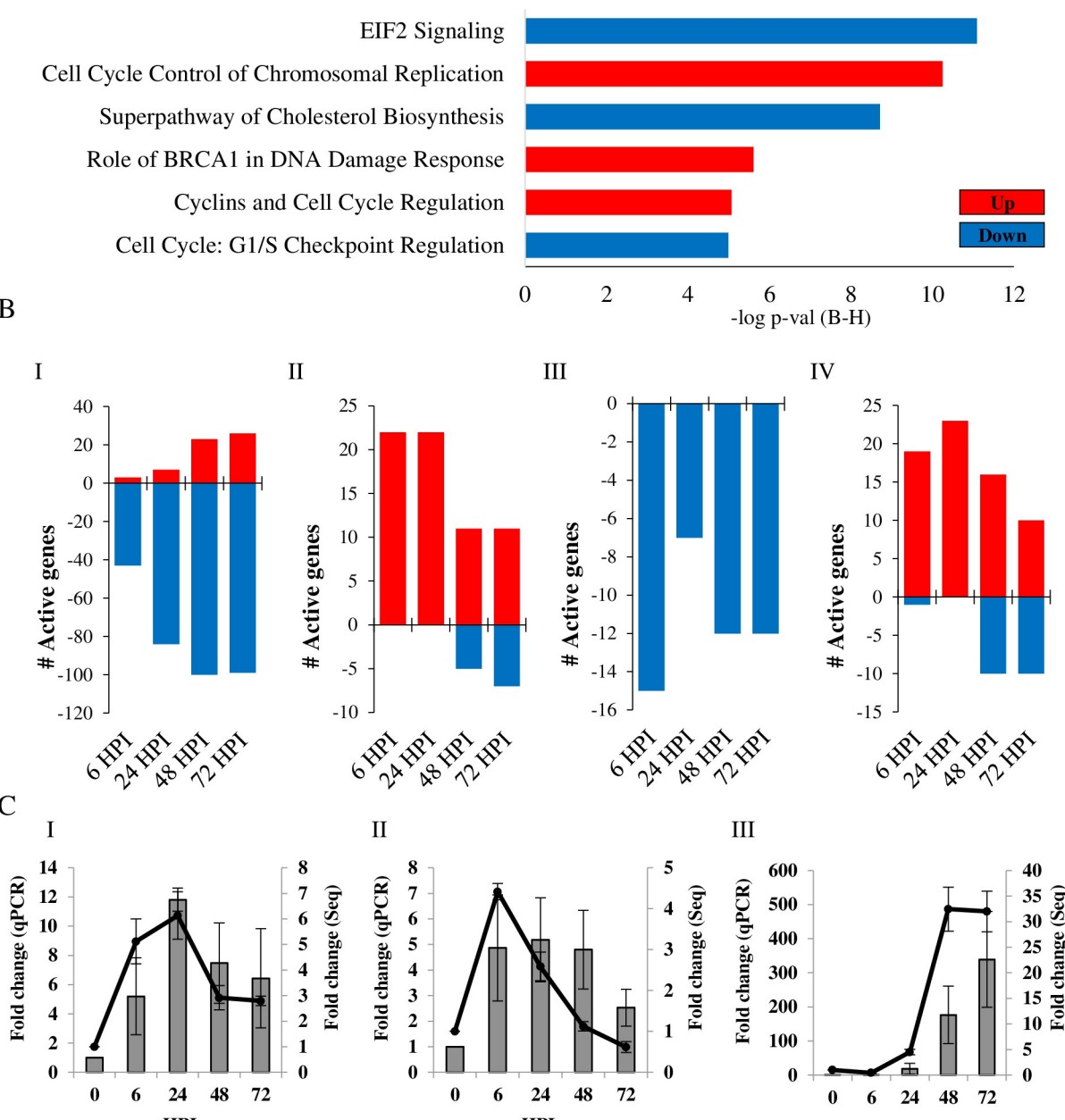

**Fig 2. Analysis of canonical pathways in HCoV-229E-infected cells.** (A) Canonical pathways enrichment analysis. Analysis was conducted for HCoV-229E-infected MRC-5 cells at 6 hours post-infection (HPI) as compared to uninfected cells. Canonical pathways with highest -log p-value (using Benjamini-Hochberg correction) are presented. (B) The number of activated genes was calculated for the eIF2 (I), cell cycle control of chromosomal replication (II), cholesterol biosynthesis (III) and role of BRCA1 in DNA damage response (IV) pathways. Up- and downregulated pathways are presented in red and blue, respectively. (C) Validation of gene expression. Cells were infected with HCoV-229E and total RNA was extracted and treated as described in the Methods section. Relative gene expression was calculated for *ORC1* (I), *CDK1* (II) mRNA (belong to cell cycle control of chromosomal replication pathway) and *ATF3* (III) mRNA (belongs to eIF2 pathway). Fold-change in transcripts levels, as measured by qPCR analysis (Bars, left Y axis), was compared to the results of the NGS analysis (line, right Y axis).

cells, 15 of which at 6 HPI (Fig 2B-III). During infection, the BRCA1 and DNA damage response pathway was active (48.75% of 80 genes) in MRC-5 cells. Many genes in the pathway were upregulated within 6–24 HPI, followed by a gradual decline in gene expression as infection persisted (Fig 2B-IV). To confirm the data of RNA-Sequencing, MRC-5 cells were infected with HCoV-229E and total RNA was extracted and analyzed by qRT-PCR for the detection of several selected mRNAs with differential expression (Fig 2C). Relative gene expression was calculated for 2 transcripts in the cell cycle control of chromosomal replication pathway, *ORC1* and *CDK1* mRNA (Fig 2C-I and II respectively) and one eIF2 pathway factor, *ATF3* mRNA (Fig 2C-III). We found that the fold change from qPCR (Fig 2C) is correlated to fold change of from NGS analysis for these genes. Next, expression of genes involved in antiviral response were analyzed. In a database of 319 genes involved in "antiviral response", 105 were active in MRC-5 cells during HCoV-229E infection. Strikingly, as shown in Fig 3A, 82% of the antiviral genes were downregulated at 6 HPI, and 65% were downregulated genes at 24 HPI. Only at 48–72 HPI did the profile change, with 58% of the genes showing upregulated

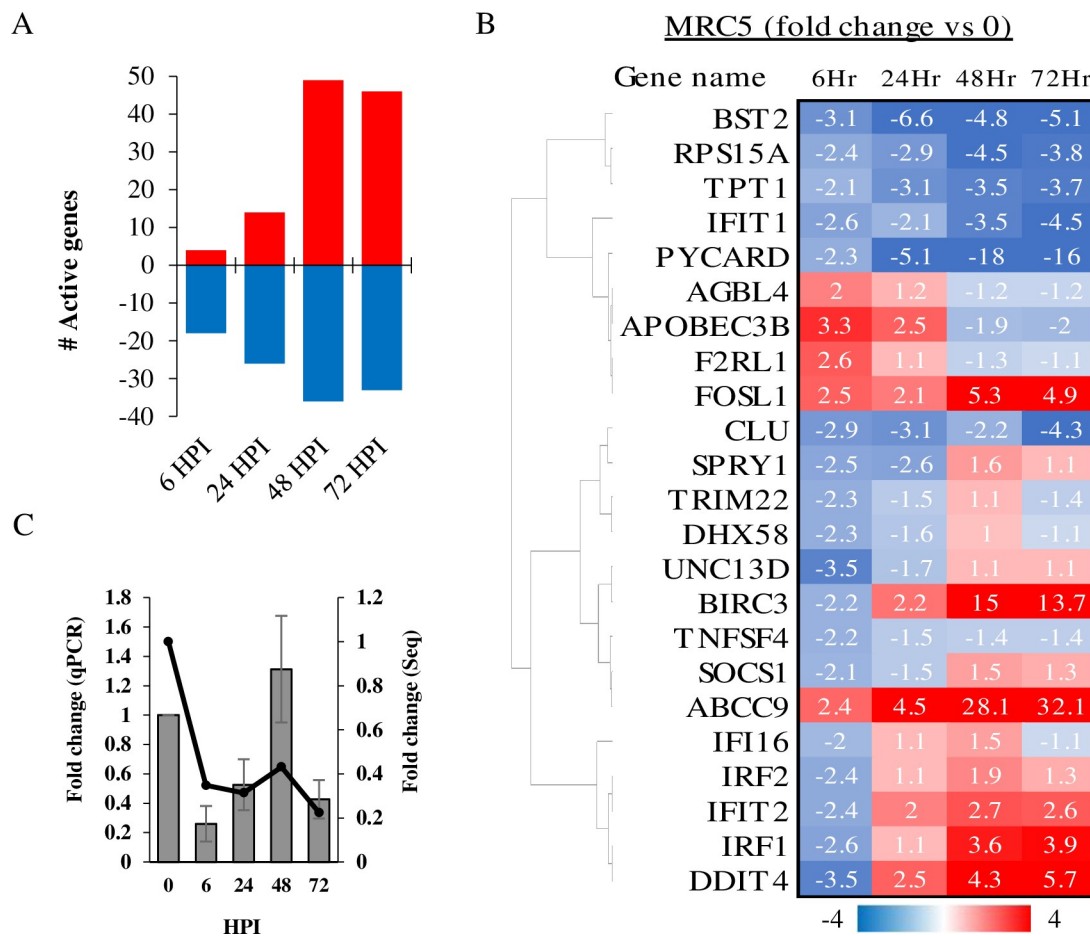

**Fig 3. Expression of genes involved in antiviral response following HCoV-229E infection.** (A) Sum of up and downregulated genes in MRC-5 cells at each time point post HCoV-229E infection are presented in red and blue, respectively. (B) Early activated antiviral response genes. Genes that were >2 or <-2 fold change in MRC-5 cells at 6 hours post HCoV-229E infection are presented in fold change heat map in following HPI. (C) Validation of gene expression. Cells were infected with HCoV-229E and total RNA was extracted and treated as described in the Methods section. Relative gene expression was calculated for *CLU* mRNA. Fold-change in transcripts levels, as measured by qPCR analysis (Bars, left Y axis), was compared to the results of the NGS analysis (line, right Y axis).

levels as compared to control cells (Fig 3A). The temporal expression of 23 early activated anti-viral genes (6 HPI), as shown in Fig 3B, indicates that most of the downregulated genes remained inactive (below 2 fold change) during infection. In general, these genes included a cluster of ISGs, e.g., the transcription factors interferon regulatory factor 1 and 2 (IRF1 and IRF2) and interferon-induced protein with tetratricopeptide repeats (IFITs) family members, a major branch of the immune antiviral response [21, 22]. Furthermore, an early regulator of the response to viral RNA detection, DHX58 (also known as LPG2 or RIG-I-like receptor 3 (RLR-3)), was downregulated (-2.25 fold) following HCoV-229E infection, as was suppressor of cytokine signaling 1 (SOCS1) (-2.09 fold). SOCS1 is involved in negative regulation of cyto-kines that signal through the JAK/STAT pathway which induces expression of IFN-stimulated genes [23, 24]. PCR analysis of *CLU* transcript, one of the early antiviral response genes (Fig 3B), validated the trend of expression observed in the NGS analysis (Fig 3C).

Apolipoprotein B mRNA editing enzyme catalytic subunit 3*B (APOBEC3B)* was upregu-lated in early antiviral responses in infected MRC-5 cells (Fig 3B). APOBEC3 proteins play essential roles in innate immunity by deaminating cytidine in RNA (C-to-U) and/or DNA (C-to-T), restricting foreign viral RNA/DNA [25]. Mutational analysis suggests that pandemic coronaviruses show a C-to-T signature of *APOBEC* activity in infected cells [26]. Thus, we ana-lyzed genomic sequences of HCoV-229E and of the most recent outbreak of SARS-CoV-2 in search of variant viruses showing characteristics of APOBEC editing activity. We downloaded 303 partial and complete genomes of HCoV-229E obtained from clinical samples (From the ViPR database) and showed that C-to-T mutations occurred as frequently as T-to-C mutations (Fig 4A), meaning that there is no strong evidence of APOBEC editing activity in the database sequences of this specific HCoV strain. However, a first analysis of broncho-alveolar lavage flu-ids derived from two Wuhan patients showed nucleotide changes that may be signatures of C-to-U RNA editing [26]. In a larger analysis, using the Global Initiative for Sharing All Influenza Data GISAID SARS-CoV-2 sequence database, Korber et al. found a distinct bias for C-to-T mutations (47%) in the spike protein [27]. In order to further characterize the mutation in the whole genomic sequence of the SARS-CoV-2, 16600 SARS-CoV-2 sequences from different origins were downloaded from the hCoV-19 database and compared; 41% of the variation between the sequences were C-to-T (Fig 4B). Higher correlation was found between C-to-T occurrences and geographic regions characterized with high mortality as compared to lower mortality (R2 = 0.222, R2 = 0.065 respectively) (Fig 4C and 4D), suggesting that the new emerging CoV may use the APOBEC nucleotide editing activity to increase its virulence.

## Discussion

This study sought to shed light on host transcriptome responses to common infecting HCoVs, using NGS and bioinformatics tools. More specifically, it aimed to examine major transcrip-tomic changes occurring over several cycles of HCoV-229E infection in human host cells. Sev-eral cellular pathways that were affected early in infection were identified, i.e., eIF2 signaling, cell cycle, cholesterol biosynthesis, BRCA1-associated DNA damage response and cellular innate immune factors.

The fluidity of membranes is dependent on the presence and structure of specific lipids and sterols. While cholesterol increases membrane fluidity by interfering with the packing of acyl chains, it can also reduce the flexibility of neighboring unsaturated acyl chains [28, 29]. Viruses may remodel lipid metabolism and membrane structure to form a microenvironment suitable for infection [29]. Lipidomic analyses have uncovered a striking rearrangement of fatty acids and phospholipids in the cellular membrane upon HCoV-229E and MERS-CoV infection [30, 31]. The current NGS analysis indicated that the cholesterol biosynthesis pathway is inhibited

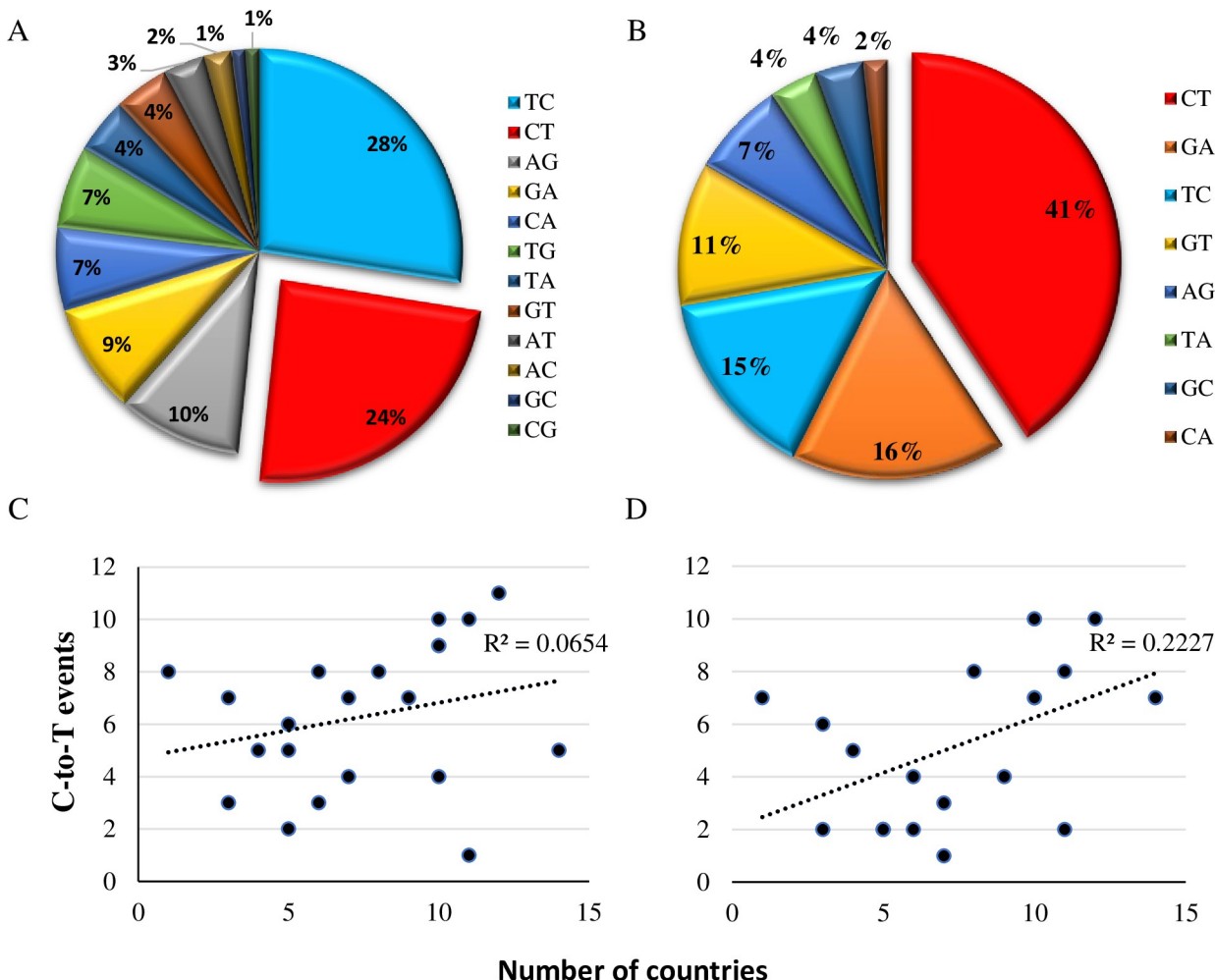

**Fig 4. APOBEC mutations in HCoV-229E and SARS-CoV-2.** (A-B) Mutation analysis of CoVs genomes. Percentage of all types of nucleotide transitions (mutation frequency >0.1) are presented in pie charts for HCoV-229E (A) and SARS-CoV-2 (B). (C-D) Correlation between independently occurring C-to-T events and the number of countries with mutations at the specific site. (C) Correlation in countries with moderate SARS-CoV-2 mortality (<99 deaths per million). (D) Correlation in countries with high SARS-CoV-2 mortality (> 99 deaths per million). Y-axis indicates the number of C-to-T alterations identified as independent events on the phylogenetic tree analysis. X-axis indicates the number of countries in which the event was identified in at least in one corona sample.

in multiple steps both at early and late stages following HCoV-229E infection (Fig 2A and 2B-III). Cholesterol can promote coronavirus murine hepatitis virus (MHV) entry to the cells [29]. When cholesterol was depleted by methyl β-cyclodextrin (MβCD) or supplemented in growth media, it reduced or increased cells susceptibility to MHV respectively [32]. These various cholesterol levels are correlated with S-protein-mediated membrane fusion rather than binding activities [32]. HCoV-229E binds to CD13 in lipid rafts and enters the cell through caveolae [33]. Depletion of plasma membrane cholesterol by MβCD disrupted the co-localization of HCoV-229E and caveolin-1 and decreased the intracellular occurrence of HCoV-229E [33]. Both SARS-CoV-1 and SARS-CoV-2 S proteins bind the host cell receptor ACE2 [34]. This receptor is localized in cholesterol-rich microdomains, which are important for the efficient interaction of the S protein with ACE2 [35]. Depletion of plasma membrane cholesterol with MβCD in Vero E6 cells relocalized the raft-resident marker caveolin-1 as well as ACE2 to a non-raft environment, and inhibited infectivity of the virus [36]. Intriguingly, cholesterol

depletion by pretreatment of Vero E6 cells with MβCD reduced the release of SARS-CoV particles and the levels of viral mRNAs were significantly decreased. However, cholesterol depletion at the post-entry stage (3 HPI) caused only a limited effect on virus particle release [37]. It remains to be determined if inhibition of the cholesterol biosynthesis superpathway in MRC-5 cells is an attempt of the host to interfere with HCoV-229E infection, or a response of the virus to establish a suitable environment for infection. If the cholesterol biosynthesis superpathway is inhibited, the basic acetyl-CoA molecule could be directed to other metabolic or synthetic pathways.

Acetyl-CoA is a major substrate used in the transfer of acetyl groups to lysine residues of histones, which results in increased accessibility of chromatin to DNA-binding proteins to the exposed sites, where they can then activate gene transcription [38]. In HCoV-229E-infected MRC-5 cells, changes in several pathways that could be linked together were observed, i.e., cell cycle regulation, control of chromosomal replication and the role of BRCA1 in DNA damage response. Cell cycle control of the chromosomal replication pathway refers to the strict control of the chromosomal origin of replication. This pathway includes the six-subunit origin recognition complex (ORC), cell division control protein 6 (Cdc6) AAA+ ATPase, minichromosome maintenance (MCM) protein complex, cyclin-dependent protein kinases (CDKs) and others [39]. Marked upregulation of this pathway was observed in early stages of HCoV-229E infection. Exploitation of host ORC has been extensively described in Epstein–Barr virus (EBV) infection, where the virus utilizes its own EBNA1 to recruit host replication initiation machinery [40, 41]. Similarly, coronaviruses manipulate the cell at various stages, which can differ between the specific coronaviruses and even between host cell types [10]. In the present analysis, upregulation of cyclins and cell cycle regulation pathway was observed, together with downregulation of checkpoint regulation Growth 1 (G1) to Synthesis (S) phases, implying that cells enter the DNA synthesis phase. Functional annotation analysis supported these observations, where "Interphase", "Segregation of chromosomes", "G1/S phase transition", "S phase" and "M phase" annotations were upregulated at 6 HPI in MRC-5 cells (data not presented, IPA software). Many DNA and RNA viruses are known to manipulate cell cycle at different stages, in efforts to increase the efficiency of virus replication by subverting host-cell functions [42]. This action is not restricted to DNA viruses or retroviruses, which primarily replicate in the nucleus, but also RNA viruses, which mainly replicate in the cytoplasm [43]. For instance, infection with measles, a negative ssRNA viruses, leads to cell arrest in G0 stage [44] and simian parainfluenza virus 5 (SV5) promotes prolonged G1 and G2 phases [45]. The positive-ssRNA avian coronavirus infectious bronchitis virus (IBV) was shown to induce cell cycle arrest at both S and G2/M phases for the enhancement of viral replication and progeny production [46–48]. Betacoronavirus MHV infection causes decreased hyperphosphorylation of the retinoblastoma protein (pRb), resulting in arrest of cell cycle progression from G1 to S [49]. Transient expression of the SARS-CoV 3a structural protein inhibits cell growth and prevents 5-bromodeoxyuridine incorporation, suggesting that it inhibits cell cycle progression [50]. The current analysis suggests that HCoV-229E causes cell cycle arrest mainly in S phase that could benefits from this cellular recruitment for chromosomal replication.

As MRC-5 cells enter DNA replication stage during HCoV-229E infection, mutations can occur. Accordingly, upregulation of BRCA1-associated DNA damage response was observed. To avoid passage of accumulated DNA damage to daughter cells, cell cycle checkpoints are activated to regulate cell growth and division, allowing cells to repair the damage before resuming cell cycle progression [51]. DNA damage activates the kinases ATM, ATR and CHK2, which, in turn, activate BRCA1 by phosphorylation [52]. The tumor repressor BRCA1 is recruited to DNA damage sites, promotes DNA damage checkpoint activation and DNA damage repair [51, 53]. BRCA1 has been suggested to be responsible for or involved in the

activation for G1/S, S-phase and G2/M checkpoints [53–56]. Activation of the ATR-dependent cellular DNA damage response is one of the mechanisms exploited by IBV to induce cell cycle S-phase arrest [46]. Activation of the cellular DNA damage response by the gammacoronavirus infectious bronchitis virus (IBV) causes cell cycle arrest in the S phase, thereby upregulating DNA replication factors in the cytoplasm, which are exploited for viral replication [10, 46].

Translation is a key step in the regulation of gene expression and one of the most energy-consuming processes in the cell. In homeostasis, the eIF2 machinery actively participates in the regulation of protein synthesis and its dysregulation is thought to contribute to pathological conditions [57]. While proper synthesis is crucial for the innate antiviral immune response, viruses rely on the host translation machinery and directly interfere with the function of transcription factors or with signaling pathways that regulate translation, to favor synthesis of viral proteins [12]. Coronaviruses modulate host protein synthesis through different mechanisms, which contributes to their pathogenicity [10]. Like other viruses, coronaviruses overload the ER with the production of viral proteins, thus triggering an ER stress response. SARS-CoV, MHV, IBV and TGEV intervene with ER stress and eIF2 regulators to ensure continuous production of functional viral proteins [13]. In the current study, downregulation of the eIF2 signaling pathway occurred at early stages of HCoV-229E infection. This is assume to be in response to the robust growth of HCoV-229E inside the cells (Fig 1A), which presumably causes ER overload at early stages of infection, thereby triggering a stress response, including manipulation of eIF2 signaling (Fig 2B-I), as reported for other CoVs [13].

Viral dsRNA and 5′-triphosphate-RNA molecules are present in the cytosol and recognized by intracellular sensors. These sensors activate signaling pathways that generally activate IFN-regulatory factors (IRFs) and NF-κB. The latter triggers the excretion of cytokines and Type-I IFN, which activate JAK-STAT signaling, which, in turn, induces expression of antiviral ISGs [58, 59]. Since the present NGS analysis identified broad downregulation of ISG transcripts (Fig 3), it can be assumed that the HCoV infection interfered with upstream regulators of IFN transcription. Indeed, downregulation of IRF1 and IRF2 transcription factors (Fig 3B), which are responsible for inducing the expression of IFNs and pro-inflammatory cytokines [60], and which are necessary for positive regulation of Toll-like receptor 3 transcription that recognizes dsRNA [22, 61–63], was observed. Similarly microarray analysis of an acute case of SARS-CoV revealed upregulation of pro- and anti-inflammatory signals and downregulation of many IFN-stimulated genes [17]. Upregulation of APOBEC3B was demonstrated in the early stages of HCoV-229E infection (Fig 3B). Induction of APOBEC3 family members is augmented by IFN-α [64]. APOBEC3B is a DNA cytidine deaminase that prevents interspecies DNA transmission by conversion of cytosines of foreign DNA to uracil, leading to hypermutations [65–67]. RNA editing activity of this cytidine deaminase was first demonstrated in human immunodeficiency virus (HIV) and later in measles, mumps and respiratory syncytial virus (RSV) [68, 69]. Upregulation of several APOBEC3 transcripts was observed in cultured human airway epithelium (HAE) cells following HCoV-NL63 infection. More specifically, three APOBEC family proteins (A3C, A3F and A3H) inhibited viral infection in vitro but did not cause hypermutation of the HCoV-NL63 genome [70]. Viral RNA sequencing and mutational analysis has suggested that both APOBEC and adenosine deaminase acting on RNA (ADAR) deaminases are involved in SARS-CoV, MERS-CoV and SARS-CoV-2 genome editing [26]. The present analysis showed that the mutational imprint of APOBEC proteins is less distinct in commonly infecting HCoV-229E, which have been circulating for several decades [70] (Fig 4). The upregulation of APOBEC3B in early stages of HCoV-229E infection (Fig 3B) could be associated with the kinetics of viral copies that are found in and out of the host cells (Fig 1), however, the lake in APOBEC C-toT editing activity in HCoV-229E genome (Fig 4) may imply an evasion mechanism of the virus from this mutational mechanism. On the other

hand, APOBEC proteins may cause antigenic drift that shapes the spread of new pandemic CoVs [26] (Fig 4).

This work used NGS and bioinformatics approaches to demonstrate the influence of HCoV-229E on the host cell transcriptome. Several pathways that were altered were discussed, mainly involving alterations of the host cell cycle regulation, metabolic processes, protein synthesis machinery and cellular innate antiviral responses. This study provides important data and expands our current understanding of the pathology and treatment targets of common HCoVs and newly emerging Coronaviridae.

## Methods

### Viruses and cell cultures

HCoV-229E virus was purchased from ATCC (VR740™) along with its recommended human host cell lines MRC-5. Cells were grown in Eagle's Minimum Essential Medium (EMEM) supplemented with 10% fetal calf serum (FCS). After a 24-hour culture to 80–90% confluency, cells were infected with 0.01 multiplicity of infection (MOI) of virus. HCoV-229E was grown in a humidified incubator with 5% CO2, at 35˚C. Cells and virus were incubated for 2–7 days as per the manufacturer recommendations.

### RNA extraction and preparation

Total RNA was extracted at 0, 6, 24, 48, 72 and 96 hours post-HCoV-229E infection, using NucliSENS® easyMag®, according to the manufacturer's protocol. At each time point, three biological repeats were tested. All RNA samples were treated with RQ1 RNase-free DNaseI (Promega, USA), according to the manufacturer's protocol. RNA quality was assessed using the Agilent High Sensitivity RNA ScreenTape for TapeStation System (Agilent Technologies, Germany).

### Real-time quantitative reverse transcription–polymerase chain reaction (qRT-PCR)

Viral load in both supernatants and cells was determined as described by Dare et al. [71]. Validation of gene expression was performed using SensiFAST™ SYBR® Lo-ROX One-Step Kit, according to the manufacturer's instructions (Bioline Reagents Ltd). The primers used for validations were as follow: *ORC1*: `F-5'-AGAGCCATCCTCGCAGAGTT, R-5'-GCAGTCCCTCC ATTCTGCAC`; *CLU*: `F-5'-GAAGAGCGCAAGACACTGCT, R-5'-CAGACGCGTGCGTAGAA CTT`; *CDK1*: `F-5'-TTGGCCTTGCCAGAGCTTTT, R-5'-CGAGCTGACCCCAGCAATAC`. *ATF3*: `F-5'-CGGAGTGCCTGCAGAAAGAG, R-5'-GTGGGCCGATGAAGGTTGAG`. Relative gene expression was calculated using the $2^{-\Delta\Delta Ct}$ method [72].

### RNA preparation, sequencing and data analysis

For NGS analysis the RNA samples (>8 RNA integrity number (RIN)) were prepared using the Illumina TruSeq RNA Library Preparation Kit v2 and sequenced in Illumina HiSeq 2500 Sequencer. Software and applications used for sequencing and data analysis were as follows: library quality control: FASTQC version 0.11.5, quality and adapter trimming: trim_galore (uses cutadapt version 1.10), mapping: Tophat2 version 2.1.0 (uses Bowtie2 version 2.2.6), gene counting: HTseq-count version 0.6.1/0.11.2, normalization and differential expression analysis: DESeq2 R package version 1.18.1. Sequencing and quality control were conducted by the Technion Genome Center (TGC). Data analysis was performed using QIAGEN Ingenuity Pathway Analysis (QIAGEN IPA) software. Transcriptomic reads at each time point were

compared to the uninfected cells. A transcript is considered active or inactive if it significantly upregulated >2 fold-change or downregulated <-2 fold-change. Canonical pathways were analyzed with highest–log(p-value) (significance of gene enrichment for any given pathway) and highest z-score (directional effect). Diseases and Functions Analysis in IPA software predicts cellular processes and biological functions based on gene expression and predicts directional change on that effect (IPA manual). A database of 319 "antiviral response"–annotated genes were pooled from the IPA database and compared to the collected transcriptome dataset. TIBCO Spotfire version 7.7.0.39 was used for data analysis, filtering and visualization, including heatmap generation. Whole genomic sequences of the SARS-CoV-2 were downloaded from the hCoV-19 data base (GSAID; more than 16600 sequences, at May 12 2020). Sequences of HCoV-229E were obtained from the NIAID Virus Pathogen Database and Analysis Resource (ViPR) [73] through the web site at http://www.viprbrc.org/. For genomic analysis we used the bioinformatics toolkit Augur (https://nextstrain-augur.readthedocs.io/en/stable/).

This work was performed in partial fulfillment of the requirements for a Ph.D. degree of Nehemya Friedman, Sackler Faculty of Medicine, Tel Aviv University, Israel.

## Author Contributions

**Conceptualization:** Michal Mandelboim.

**Formal analysis:** Michal Mandelboim.

**Investigation:** Nehemya Friedman, Yaron Drori, Nitzan Kol, Omri Nayshool, Michal Mandelboim.

**Methodology:** Nehemya Friedman, Jasmine Jacob-Hirsch, Eyal Eran.

**Resources:** Michal Mandelboim.

**Supervision:** Ella Mendelson, Gideon Rechavi.

**Writing – original draft:** Nehemya Friedman.

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
