## [Decision Letter · Decision Letter 0]

4 Jan 2021

PONE-D-20-36572

Transcriptomic profiling and genomic mutational analysis of Human corona virus (HCoV)-229E -infected human cells

PLOS ONE

Dear Dr. Mandelboim,

Thank you for submitting your manuscript to PLOS ONE. After careful consideration, we feel that it has merit but does not fully meet PLOS ONE’s publication criteria as it currently stands. Therefore, we invite you to submit a revised version of the manuscript that addresses the points raised during the review process.

We look forward to receiving your revised manuscript.

Kind regards,

Binod Kumar, PhD

Academic Editor

PLOS ONE

Journal Requirements:

2.Thank you for stating the following financial disclosure:

Reviewers' comments:

Reviewer's Responses to Questions

**Comments to the Author**

1. Is the manuscript technically sound, and do the data support the conclusions?

Reviewer #1: Yes

Reviewer #2: Yes

2. Has the statistical analysis been performed appropriately and rigorously? 

Reviewer #1: No

Reviewer #2: I Don't Know

3. Have the authors made all data underlying the findings in their manuscript fully available?

Reviewer #1: Yes

Reviewer #2: Yes

4. Is the manuscript presented in an intelligible fashion and written in standard English?

Reviewer #1: Yes

Reviewer #2: Yes

5. Review Comments to the Author

Reviewer #1: Title and other places in the manuscript: Please change 'corona virus' to 'coronavirus'

Abstract: HCoV-229E favors S phase for viral infection - Not enough data to substantiate this claim

Line 87- change 'MCR5' to 'MCR-5' - also throughout the manuscript

Line 88- hours post infection (hpi)

Line 88- change several days to 4 days

Line 89- It is important to clearly mention here that viral titers were estimated both from the (A) infected cells as well as the from (B) culture supernatants

Line 90- mention here that the canonical pathway analysis is done for the 6 hpi timepoint. Include some background, or even a little methodological detail here.

Line 99- What parameter do you use to distinguish between active and inactive genes? Fig 2B shows only active genes

Line 101- 'which are coding to' replace by 'which code for'

Line 105- but were later less active?

Line 106- mention here that the fold change from qPCR is correlated to fold change from NGS (include correlation coefficient if applicable)

Line 118- The genes not identified- are some of them not present in MCR-5 cells?

Line 123- Do you (and how) define inactive genes from the NGS analysis? Or only from qPCR data?

Line 135- Mention the pattern

Line 139- Does this mean that there is no evidence of editing in the database sequences?

Line 140- 'that may be signatures of RNA editing' signatures of C-to-T RNA editing?

Line 145- Since the database analysis suggests that C-to-T editing by APOBEC is more characteristic of SARS-CoV-2 rather than HCoV-229E, is this a strain specific effect observed here?

Line 206- data to substantiate this claim?

Line 375 and 377- Journal details missing

Line 481- and incubated

Reviewer #2: This is a very useful Research Article providing insights in alterations of transcriptomics in MRC5 cells that have been infected with HCoV-229E.

Provided data and their discussion are intelligible and interesting. The conception of the conducted research is original and helpful.

Appropriated controls are used and conclusions are based on the presented data and the annotated bibliography.

All data are fully available and well described in the figures.

I would like to indicate some possible additional considerations, for example at line 161 it might be useful to commend on the cholesterol biosynthesis pathway being downregulated both at early and late stages following HCoV-229E infection.

Could the upregulation of APOBEC3B (as antiviral reflection) in early stages of HCoV-229E infection be associated with the kinetics of viral copies that are found in and out of the host cells?

The Research Article is a thorough effort to document transcriptomic changes after HCoV-229E infection, that is being considered a "benign" infection.

6. PLOS authors have the option to publish the peer review history of their article (what does this mean?). If published, this will include your full peer review and any attached files.

Reviewer #1: No

Reviewer #2: No

---

## [Author Response · Author response to Decision Letter 0]

28 Jan 2021

Response to Reviewers

Our sincere gratitude for your dedication and diligent work! We thank for the opportunity to learn from your experience and your constructive comments.

Reviewer #1

Comment 1

Title and other places in the manuscript: Please change 'corona virus' to 'coronavirus'

Answer 1

'Corona virus' in the title and in line 135 were corrected to 'coronavirus'

Comment 2

Abstract: HCoV-229E favors S phase for viral infection - Not enough data to substantiate this claim

Answer 2

Indeed, this is a statement that is not supported by cell cycle analysis and should rather be phrased as an assumption or suggestion. Thus, we added the word 'may' to the sentence: "…suggest that HCoV-229E may favors S phase for viral infection" . 

Comment 3

Line 87- change 'MCR5' to 'MCR-5' - also throughout the manuscript

Answer 3

All 23 "MRC5" were changed to "MCR-5".

Comment 4

Line 88- hours post infection (hpi)

Answer 4

Certainly, it is not sufficient that the meaning of this acronym appears only in the methods and it also must be at the first time it appears in the results.

Comment 5

Line 88- change several days to 4 days

Answer 5

Corrected. 

Comment 6

Line 89- It is important to clearly mention here that viral titers were estimated both from the (A) infected cells as well as the from (B) culture supernatants

Answer 6

Sentences were rephrased to emphasize this description: 

"RNA was extracted at various time points following infection of MRC-5 with HCoV-229E and viral titers were estimated both from the infected cells as well as the from culture supernatants by qPCR analysis (Fig 1A-B). Rapid viral propagation inside the host cells, reaches maximal levels at 24 hours post infection (HPI) and remained stable for 4 days (Fig 1A)."

Comment 7

Line 90- mention here that the canonical pathway analysis is done for the 6 hpi timepoint. Include some background, or even a little methodological detail here.

Answer 7

The canonical pathway analysis was done for all time points. We focused at 6 HPI only on early activated antiviral genes (Figure 3B) (lines 121-123)

Comment 8

Line 99- What parameter do you use to distinguish between active and inactive genes? Fig 2B shows only active genes

Answer 8

The parameter is of active/inactive transcript is being significantly >2 upregulated or <-2 downregulated in fold-change as compared to uninfected cells. We added this information in the text next to the sentence in line 99 and in the methods. Thank you for emphasizing this point. 

Comment 9

Line 101- 'which are coding to' replace by 'which code for'

Answer 9

This sentence was corrected.

Comment 10

Line 105- but were later less active?

Answer 10

Thank you for this comment. You are correct. We sorry for using an inappropriate term and replaced the word "lightly" to "were later less". 

Comment 11

Line 106- mention here that the fold change from qPCR is correlated to fold change from NGS (include correlation coefficient if applicable)

Answer 11

Thank you for this comment. It is a very important remark and we added it to the manuscript. 

Comment 12

Line 118- The genes not identified- are some of them not present in MCR-5 cells?

Answer 12

Indeed, some genes may be present in immune cells however some of these genes may have low copies number, thus had nonsignificant p-value and were excluded from downstream analysis. Mentioning the "256 genes" does not contribute to the discussion and may even be misleading, thus we deleted it.

Comment 13

Line 123- Do you (and how) define inactive genes from the NGS analysis? Or only from qPCR data?

Answer 13

See comment/answer 8.

All the data and calculations were performed on NGS analysis. qPCR was used only to validated gene expression of selected genes. 

Comment 14

Line 135- Mention the pattern

Answer 14

Since we wanted to quantify point mutations in the virus genome and avoided the calculation of mutation patterns, we rephrased the citation from "show a pattern of APOBEC activity" to "show a C-to-T signature of APOBEC activity".

Comment 15

Line 139- Does this mean that there is no evidence of editing in the database sequences?

Answer 15

Thanks for your remark. Indeed, this conclusion should be written here. We added the following: "…C-to-T mutations occurred as frequently as T-to-C mutations (Fig 4A), meaning that there is no strong evidence of APOBEC editing activity in the database sequences of this specific HCoV strain."

Comment 16

Line 145- Since the database analysis suggests that C-to-T editing by APOBEC is more characteristic of SARS-CoV-2 rather than HCoV-229E, is this a strain specific effect observed here?

Answer 16

Evidences on C-to-T editing in SARS-CoV, MERS-CoV and SARS-CoV-2 are supported in other studies that were cited in this article. As regarding HCoV-229E, we analyzed only HCoV-229E. We added a clarification in line 139 (answer to comment 14)

Comment 17

Line 140- 'that may be signatures of RNA editing' signatures of C-to-T RNA editing?

Answer 17

This is better describing the quoted study and the sentence was corrected..

Comment 18

Line 206- data to substantiate this claim?

Answer 18

Since no cell cycle analysis was performed in this study, we could give only a suggestion rather than a claim or a fact. The data is shone in figure 2A and more specifically in lines 191-195:

"In the present analysis, upregulation of cyclins and cell cycle regulation pathway was observed, together with downregulation of checkpoint regulation Growth 1 (G1) to Synthesis (S) phases, implying that cells enter the DNA synthesis phase. Functional annotation analysis supported these observations, where "Interphase", "Segregation of chromosomes", "G1/S phase transition", "S phase" and "M phase" annotations were upregulated at 6 HPI in MRC-5 cells (data not presented, IPA software)."

Comment 19

Line 375 and 377- Journal details missing

Answer 19

Citations were corrected. The study of Korber B et.al (citation #27) was a preprint version in biorxiv and we replaced it with an updated journal version that was published in Cell.

Comment 20

Line 481- and incubated

Answer 20

An unfortunate mistake that was corrected. Thank you..

Reviewer #2 

Comment 1

at line 161 it might be useful to commend on the cholesterol biosynthesis pathway being downregulated both at early and late stages following HCoV-229E infection.

Answer 1

Thank you for emphasize these details. We added this sentence to the discussion

Comment 2

Could the upregulation of APOBEC3B (as antiviral reflection) in early stages of HCoV-229E infection be associated with the kinetics of viral copies that are found in and out of the host cells?

Answer 2

This is a very interesting point for discussion! Indeed APOBEC, as ISG could be associated with the kinetics of HCoV-229E, however, the lake in APOBEC C-toT editing activity in HCoV-229E genome may imply an evasion mechanism of the virus from this mutational mechanism.

---

## [Decision Letter · Decision Letter 1]

2 Feb 2021

Transcriptomic profiling and genomic mutational analysis of Human coronavirus (HCoV)-229E -infected human cells

PONE-D-20-36572R1

Dear Dr. Mandelboim,

We’re pleased to inform you that your manuscript has been judged scientifically suitable for publication and will be formally accepted for publication once it meets all outstanding technical requirements.

Kind regards,

Binod Kumar, PhD

Academic Editor

PLOS ONE

Additional Editor Comments (optional):

Reviewers' comments:

Reviewer's Responses to Questions

**Comments to the Author**

1. If the authors have adequately addressed your comments raised in a previous round of review and you feel that this manuscript is now acceptable for publication, you may indicate that here to bypass the “Comments to the Author” section, enter your conflict of interest statement in the “Confidential to Editor” section, and submit your "Accept" recommendation.

Reviewer #1: All comments have been addressed

2. Is the manuscript technically sound, and do the data support the conclusions?

Reviewer #1: Yes

3. Has the statistical analysis been performed appropriately and rigorously? 

Reviewer #1: Yes

4. Have the authors made all data underlying the findings in their manuscript fully available?

Reviewer #1: Yes

5. Is the manuscript presented in an intelligible fashion and written in standard English?

Reviewer #1: Yes

6. Review Comments to the Author

Reviewer #1: Thanks for addressing all the comments and concerns raised earlier. The manuscript in the present format is acceptable for publication in PLoS One.

7. PLOS authors have the option to publish the peer review history of their article (what does this mean?). If published, this will include your full peer review and any attached files.

Reviewer #1: No

---

## [Editor Report · Acceptance letter]

17 Feb 2021

PONE-D-20-36572R1 

Transcriptomic profiling and genomic mutational analysis of Human coronavirus (HCoV)-229E -infected human cells 

Dear Dr. Mandelboim:

I'm pleased to inform you that your manuscript has been deemed suitable for publication in PLOS ONE. Congratulations! Your manuscript is now with our production department. 

Kind regards, 

on behalf of

Dr. Binod Kumar 

Academic Editor

PLOS ONE